# Clinical Experience with Abemaciclib in Patients Previously Treated with Another CDK 4/6 Inhibitor in a Tertiary Hospital: A Case Series Study

**DOI:** 10.3390/cancers15184452

**Published:** 2023-09-07

**Authors:** Alicia Milagros de Luna Aguilar, Javier David Benitez Fuentes, Justo Ortega Anselmi, Jennifer Olalla Inoa, Paloma Flores Navarro, Alfonso Lopez de Sá, Jesus Fuentes Antras, Cristina Rodríguez Rey, Aída Ortega Candil, Fernando Moreno Antón, Jose Ángel García Sáenz

**Affiliations:** 1Department of Medical Oncology, Hospital Clinico San Carlos, IdISSC, Calle Profesor Martín Lagos, S/N, 28040 Madrid, Spain; javier.fuentes@wales.nhs.uk (J.D.B.F.); jennifer.olalla@marinasalud.es (J.O.I.); palomaalejandra.flores@salud.madrid.org (P.F.N.); fmorenoa@salud.madrid.org (F.M.A.); jgsaenz@salud.madrid.org (J.Á.G.S.); 2Velindre Cancer Centre, Velindre University NHS Trust, Cardiff CF14 2TL, UK; 3Department of Nuclear Medicine, Hospital Clinico San Carlos, IdISSC, Calle Profesor Martín Lagos, S/N, 28040 Madrid, Spain; cristina.rodriguez.rey@gmail.com (C.R.R.);

**Keywords:** metastatic breast cancer, continuing CDK4/6 inhibition, abemaciclib, endocrine therapy, rechallenge

## Abstract

**Simple Summary:**

This research aimed to study the effectiveness and safety of a drug called abemaciclib in patients with metastatic breast cancer (MBC) who had previously received similar treatments. The medical records of 11 patients who were given abemaciclib after their disease worsened despite prior therapies were analyzed. The results demonstrated that abemaciclib, alone or in combination with tamoxifen, showed positive effects against MBC. On average, patients experienced six months without disease progression, and some even had significant improvements, such as one patient achieving complete resolution of liver metastases. The most observed side effects, including diarrhea and fatigue, were generally mild. These findings might add further evidence that abemaciclib could be a safe and effective treatment option for MBC patients who have not responded to previous therapies. This research provides valuable insights for making informed decisions regarding future treatment approaches in the medical community.

**Abstract:**

The three approved cyclin-dependent kinase 4/6 (CDK4/6) inhibitors, including abemaciclib, have shown differences in their preclinical, pharmacological, and clinical data. Abemaciclib stands out for its broader target range and more rapid and intense activity. It has demonstrated efficacy as a monotherapy or in combination with tamoxifen in endocrine-refractory metastatic breast cancer (MBC) patients with prior chemotherapy. However, the clinical data on abemaciclib after exposure to previous CDK4/6 inhibitors are limited. In this single-center retrospective case series, we identified all patients who received abemaciclib until February 2022 after experiencing documented progression on palbociclib or ribociclib. The safety profile and clinical outcomes of abemaciclib treatment in this specific patient cohort were evaluated. Eleven patients were included in this retrospective case series, nine receiving abemaciclib with tamoxifen. Eight patients had visceral involvement, and the median age was 69 (ranging from 42 to 84). The median time from the end of prior CDK4/6 inhibitor treatment to abemaciclib initiation was 17.5 months (ranging from 3 to 41 months). Patients had undergone a median of three prior therapies (ranging from 1 to 7), including chemotherapy in 54.5% of cases. The median follow-up time was six months (ranging from 1 to 22 months). The median progression-free survival (PFS) was 8 months (95% CI 3.9–12). Five patients continued abemaciclib treatment, and one patient with liver metastases achieved a complete hepatic response. The most common adverse events were diarrhea (72.7%, no grade ≥ 3) and asthenia (27.3%, no grade ≥ 3). Our preliminary findings suggest that abemaciclib could be an effective and safe treatment option for MBC patients who have previously received palbociclib or ribociclib.

## 1. Introduction

Breast cancer is the leading cancer diagnosed in women globally and the second most common cause of cancer-related death in women, following lung cancer [1]. Hormone receptors (HRs) are expressed in about 70% of breast cancer cases, with early-stage diagnosis being more frequent than metastatic-stage diagnosis [2].

Although estrogen-receptor-positive and epidermal growth factor receptor 2-negative (ER+/HER2-) metastatic breast cancer is considered incurable, there has been a shift in treatment strategies in recent years. CDK4/6 inhibitors, such as ribociclib, palbociclib, and abemaciclib, have emerged as the mainstay of treatment for these patients. These inhibitors work by inhibiting CDK4/CDK6-dependent phosphorylation of retinoblastoma (Rb), which blocks proliferation by inhibiting the progression of tumor cells from the G1 phase into the S phase of the cell cycle. Multiple recent studies have supported using CDK4/6 inhibitors (CDK4/6i) in combination with endocrine therapy in MBC, establishing ribociclib, palbociclib, and abemaciclib as first- and second-line therapies [3].

Abemaciclib has a different clinical and preclinical mode of action compared to other CDK4/6 inhibitors, blocking the activity of other cellular targets, such as Cyclin Dependent Kinase 2 (CDK2/Cyclin A/E) and Cyclin-Dependent Kinase 1 (CDK1/Cyclin B), which results in the arrest of the cell cycle in both the G1 and G2 phases. Additionally, abemaciclib induced senescence and apoptosis at lower concentrations and in earlier stages [4]. In a phase II clinical trial, abemaciclib demonstrated significant activity as a monotherapy or in combination with tamoxifen for patients with metastatic breast cancer who had undergone prior chemotherapy treatment and were considered endocrine-refractory [5,6].

A significant clinical issue is the emergence of resistance to these therapies [7]. Numerous cell-cycle-regulatory proteins, including RB Transcriptional Corepressor 1 (RB1), Cyclin E1 (CCNE1), Cyclin E2 (CCNE2), Cyclin-Dependent Kinase 6 (CDK6), and aurora kinase (AURKA), have been identified as key players in several mechanisms. However, other oncogenic pathways such as Erb-B2 Receptor Tyrosine Kinase 2 (ERBB2), Fibroblast Growth Factor Receptor (FGFR), AKT Serine/Threonine Kinase 1 (AKT), and RAS also appear to be crucial. The complexity of the underlying molecular processes, as evidenced by the involvement of these pathways and cell-cycle-regulatory proteins, poses significant challenges to developing effective treatments [7,8,9,10,11,12]. Ongoing research is being conducted using in silico techniques to explore the role of cyclin-dependent kinases in multiple cancers [13,14,15,16].

There are limited clinical data on abemaciclib treatment after a prior CDK 4/6i exposure. Considering the different patterns of action of abemaciclib and the potential usefulness of abemaciclib when CDK4/6 inhibitor resistance appears, this study aimed to analyze the safety and efficacy of abemaciclib in a cohort of HR+/HER2- MBC patients who had previously progressed on palbociclib or ribociclib at our institution. The primary objectives were to assess the safety profile and clinical outcomes of these patients treated with abemaciclib.

## 2. Materials and Methods

We conducted a single-center, retrospective identification of patients who had been or were being treated with abemaciclib until February 2022. The data were extracted from the hospital pharmacy’s electronic records. From this group of patients, we included those who had received abemaciclib after a previously documented progression on palbociclib or ribociclib. Medical records of patients included in the study were reviewed retrospectively. Clinical characteristics including age, sex, time since diagnosis of metastatic disease, number of previous lines, previous treatment modalities, time until treatment with abemaciclib, endocrine partner used in combination with abemaciclib, visceral and central nervous system (CNS) involvement, and tumor markers from the start of abemaciclib treatment until disease progression were summarized using standard descriptive statistics.

In this study, descriptive statistics were used to summarize and present the results. The efficacy of abemaciclib measured by clinical benefit rate (CBR) and progression-free survival (PFS) was analyzed retrospectively. CBR was defined as the percentage of all patients with a complete or partial response or stable disease at 24 weeks. PFS was defined as the time in months from the start of abemaciclib to death, disease progression, or the date of the last follow-up. The response was assessed by the local radiologist based on RECIST v1.1 [17]. Median PFS was calculated according to the Kaplan–Meier method. Statistical analysis was performed with SPSS (version 28.0.1.1 IBM SPSS Statistics). It is important to note that due to the retrospective nature of the study, the findings should be interpreted cautiously.

The safety profile of abemaciclib was analyzed by reviewing all the adverse events (AEs) documented in medical records. The severity of adverse events was graded according to CTCAE version 5.0 [18]. Grade 3 and 4 AEs were reported separately.

The study was conducted according to the guidelines of the Declaration of Helsinki and approved by the Institutional Review Board. The Hospital Clinico San Carlos Ethics Committee approved the study with the code 22/083-O_M_SP. Written informed consent was obtained from all participants included in the study. The authors affirm that human research participants provided informed consent for the publication of the images in Figures 2 and 3.

## 3. Results

### 3.1. Patients

The first patient to fulfill inclusion criteria started abemaciclib in April 2020. In February 2022, 11 patients met the inclusion criteria and were included in the study.

All the patients were women; the median age was 69 years (range 42–84 years), and most of them were Caucasian (90.9%). All the tumors were ER+/HER2- MBC. Median time to abemaciclib from the end of previous CDK4/6 inhibitors was 17.5 months (range 3–41 months), and the median number of therapies until abemaciclib use was three (range 1–7), including chemotherapy in 54.5% of patients.

Considering the combined therapy used with abemaciclib, 9 out of 11 received tamoxifen. Regarding disease extension, eight patients had visceral involvement, and none had brain metastasis.

Baseline patient and disease characteristics are presented in Table 1.

### 3.2. Efficacy

The median follow-up time was six months (range 1–22 months). At the time of data analysis, five patients continued abemaciclib. Median PFS was 8 months (95% CI 3.9–12). The Kaplan–Meier curve for progression-free survival is presented in Figure 1. The PFS of each patient is represented in Table 2. CBR at 24 weeks was identified in five out of eight evaluable patients.

The benefit obtained on previous CDK4/6 inhibitors did not seem to determine the benefit of abemaciclib in the small sample of patients included in this retrospective single-center case series. Two patients with a prolonged PFS on prior CDK4/6 inhibitors achieved a PFS of more than six months on abemaciclib, but this trend was not maintained in the remaining patients. Considering the heterogeneity of the patients included in the study according to the different treatment strategies used and the time elapsed from prior CDK4/6 inhibitors to abemaciclib therapy, no conclusions can be drawn regarding which patients would benefit most from this strategy.

Patient 6 was a 51-year-old woman with liver metastases and three prior lines, including ribociclib in combination with an aromatase inhibitor. The benefit obtained with prior ribociclib had also been prolonged. A relevant and unexpected response was obtained in this patient, who achieved a complete hepatic response after three months on abemaciclib and tamoxifen. The benefit after seven months of treatment continued at the time of data analysis (Figure 2 and Figure 3).

### 3.3. Safety

Ten patients had at least one adverse event (AE). The most common AEs of any grade were diarrhea and asthenia. Diarrhea was experienced by eight patients (72.7%), mainly grade 1 (n = 5, 45.4%) and grade 2 (n = 3, 27.3%). Asthenia was experienced by three patients (27.3%), typically grade 1 (n = 1, 9.1%) and grade 2 (n = 2, 18.2%). No grade 3 or 4 diarrhea or asthenia was documented according to CTCAE version 5.0 [18].

Grade 2 blood creatinine increase was reported in one patient, in which no treatment adjustment was needed.

Neutropenia grade 4 was recorded in a heavily treated patient who had previous episodes of neutropenia grade 3 and grade 4 when she was on chemotherapy requiring granulocyte-colony stimulating factors (G-CSFs). During the grade 4 neutropenia episode with abemaciclib, no G-CSF was needed. Neutropenia resolved after one week off treatment, and it did not recur after one level of dose reduction.

## 4. Discussion

CDK4/6 inhibitors are the main treatment strategy in ER+/HER2- MBC [15,16]. Currently, chemotherapy is only recommended for patients experiencing visceral crises in the first-line setting [19,20]. Different studies have evaluated using palbociclib, ribociclib, and abemaciclib in these patients [21,22,23,24,25,26,27,28]. Combined with endocrine therapies in the first and second line, these agents significantly improve PFS and overall survival (OS). These therapies’ effectiveness and enhanced safety profile have enabled the postponement of more toxic treatments in this context.

Results from two studies, MONARCH-1 and nextMONARCH, have shown interesting clinical data efficacy in HR+/HER2- endocrine-refractory MBC patients in monotherapy and combination with tamoxifen [5,6].

A phase II trial called MONARCH-1 investigated the efficacy of abemaciclib 200 mg twice daily in 132 patients with HR+/HER2- endocrine-refractory MBC. These patients had received a median of three previous lines of both endocrine and chemotherapy regimens. Despite being heavily used in treatment, abemaciclib as a single agent showed an ORR of 19.7% and a median PFS of 6.0 months. In the final analysis, conducted after 18 months of follow-up, the median OS was 22.3 months [5].

The results of the NextMONARCH phase II clinical trial were published in 2021. Patients with the same characteristics as in MONARCH-1 were included. This study evaluated abemaciclib in monotherapy at two doses (A-150 and A-200) and in combination with tamoxifen (A+T). At 15.1 months of follow-up, the efficacy analysis showed that A+T had a numerically longer PFS of 9.1 months compared to A-200 with 7.4 months (HR, 0.815; *p* = 0.293), but this difference was not statistically significant. The PFS results of A-200 and A-150 showed no inferiority. The CBR was numerically different, but no statistically significant differences were observed among the three treatment arms [6].

Despite the clinical success of CDK4/6 inhibitors, emerging resistance to them has posed a new challenge in HR+/HER2+ MBC. Data support that this resistance is likely multifactorial [7,29]. Several resistance pathways have been identified that involve different cell-cycle-regulatory proteins. While high expression of CCNE2 is linked to unfavorable outcomes after endocrine therapy, amplification of CCNE1 and CCNE2 has been associated with resistance to CDK4/6 inhibitors [10,11]. C. Yang et al. conducted a study on cell lines exposed to abemaciclib and found that CDK6 amplification increased CDK6 expression and reduced response to CDK4/6 inhibitors [9]. RB1 alterations have been identified in a small fraction (3–5%) of patients in different studies and are associated with poor response to CDK4/6 inhibitors [8]. AURKA is known to regulate cell-cycle progression, and its overexpression in breast cancer is commonly associated with an ER-low/basal phenotype. However, emerging evidence suggests that AURKA also mediates resistance to CDK4/6 inhibitors in vitro and in tumor samples [29].

Other oncogenic pathways also appear to be relevant in resistance to CDK4/6 inhibitors. Previous studies suggest the implication of activating mutations in ERBB2, amplification events in FGFR1 and FGFR2, and alterations in AKT1 and RAS in the resistance to both antiestrogens and CDK4/6 inhibitors in vitro [12,29].

It is unclear if palbociclib, ribociclib, and abemaciclib are functionally equivalent in HR+/HER2+ MBC patients. Marc Hafner et al. compared the three drugs using in vitro and in vivo xenograft tumors, revealing differences. Abemaciclib was the most effective CDK4/6 inhibitor, capable of reducing pRb phosphorylation and arresting cells in the G1 phase at a lower drug concentration than ribociclib and palbociclib. Moreover, abemaciclib demonstrated activity against various kinases, including CDK1/cyclin B, CDK2/cyclin A/E, CDK7/cyclin H, and CDK9/cyclin K/T1. Due to its ability to suppress CDK1 and CDK2, which are involved in the progression through the S phase and mitosis, abemaciclib was able to arrest the cell cycle not only in G1 but also in G2 [4].

Considering the preclinical and clinical differences observed between the three CDK4/6 inhibitors approved, the hypothesis that abemaciclib could be effective after progression on palbociclib or ribociclib has emerged in recent years (4). Data have been published recently demonstrating clinical efficacy with abemaciclib in patients after progression on palbociclib or ribociclib. Wander et al. reported the first multicenter experience with 87 patients showing clinical benefit in a substantial number of patients. The median PFS was 5.3 months (95% CI 3.5–7.8) [30,31]. Another case series published by Mariotti V. et al. demonstrated a response with abemaciclib in some patients after progression on palbociclib with a median PFS of 7.0 months (range 1.8–12.1) [32]. Some other case reports and single-center experiences also suggest the clinical efficacy of abemaciclib in this setting [33,34,35].

Considering this hypothesis, the phase III study postMONARCH is open for patients who have progressed to a combination of CDK4/6 and aromatase inhibitors. This multinational study has been presented in ASCO 2022 and is planned for 122 centers. The study anticipates enrolling 350 patients randomized to fulvestrant with abemaciclib or placebo (NCT05169567) [36,37].

Other clinical trials have been designed based on the same hypothesis, investigating the effectiveness of various targeted therapies such as enobosarm and lasofoxifene in combination with abemaciclib for ER+ HER2- MBC patients who have previously experienced progression on CDK4/6 inhibitors [38,39].

Studies have also investigated the potential advantages of switching from palbociclib and ribociclib to a different endocrine therapy after experiencing disease progression with previous CDK4/6 inhibitors. The phase II study MAINTAIN was the first prospective randomized trial published in this setting. It was first presented at the ASCO Congress in June 2022 and published in May 2023. Patients were randomized to receive fulvestrant or exemestane with or without ribociclib. Most patients (84%) had received prior palbociclib. After a median follow-up of 18.2 months, a statistical benefit of 2.53 months in the primary endpoint median PFS was shown for patients randomized to ribociclib versus placebo (HR 0.56; *p* = 0.004) [40].

Two studies were conducted to investigate the role of palbociclib. The PACE trial (NCT03147287) enrolled 220 HR+/HER2 MBC patients previously treated with CDK4/6 inhibitors, and they were randomly assigned to fulvestrant, fulvestrant with palbociclib, or fulvestrant with palbociclib and avelumab. The first results presented at the San Antonio Breast Cancer Symposium in December 2022 showed no significant difference in PFS between fulvestrant and palbociclib compared to fulvestrant alone (HR 1.11, *p* = 0.62). Still, the addition of avelumab showed a numerical benefit [41]. The PALMIRA trial first analysis results (NCT03809988) were presented in ASCO 2023. In this trial, HR+/HER2- MBC patients who had received first-line endocrine therapy and palbociclib with clinical benefit (defined as a response or stable disease ≥24 weeks) or who had progressed on palbociclib in the adjuvant setting or less than 12 months following palbociclib treatment completion were randomized 2:1 to receive a different endocrine agent alone or with palbociclib. At a median follow-up of 8.7 months, no significant improvement had been seen in PFS (HR 0.8, 95% CI 0.6–1.1, *p* = 0.206) [42].

The characteristics of these studies are summarized in Table 3.

The present retrospective single-center case series offers insights into the efficacy and safety of abemaciclib following progression on prior CDK4/6 inhibitors in HR+/HER2- metastatic breast cancer patients. However, the study’s inherent limitations, including its small sample size of 11 patients, lack of a control group, heterogeneous treatment regimens, and reliance on medical records for data extraction, call for a cautious interpretation of the findings. The retrospective nature introduces potential biases that limit the generalizability of results. Consequently, prospective studies with larger, diverse cohorts and controlled designs are necessary to better understand the role of abemaciclib in this context.

## 5. Conclusions

Abemaciclib exhibits a broader spectrum of action, suggesting its potential utility in HR+/HER2- endocrine-resistant MBC patients who have experienced progression on palbociclib or ribociclib. This study adds real-world data evidence in this setting. An ongoing phase III clinical trial is currently evaluating the efficacy of abemaciclib in this context [32,33]. Moving forward, the primary focus of future research will be identifying the patient population that benefits the most from this therapeutic strategy.

## Figures and Tables

**Figure 1 cancers-15-04452-f001:**
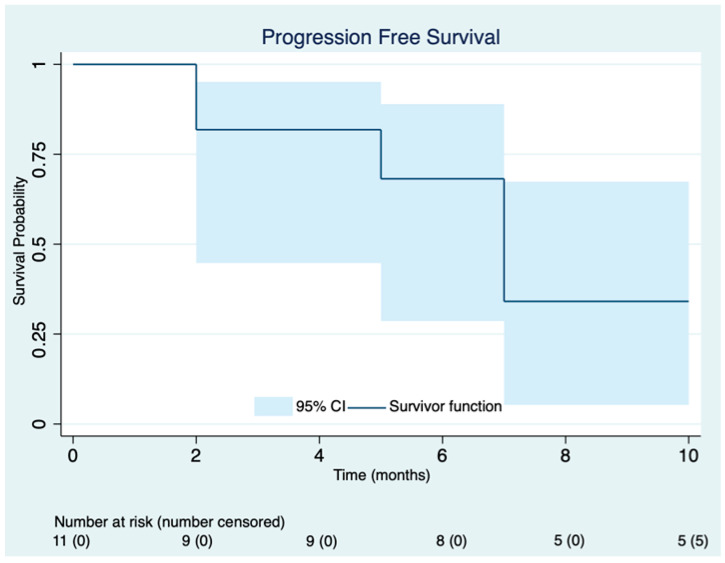
Kaplan–Meier curve for progression-free survival on abemaciclib.

**Figure 2 cancers-15-04452-f002:**
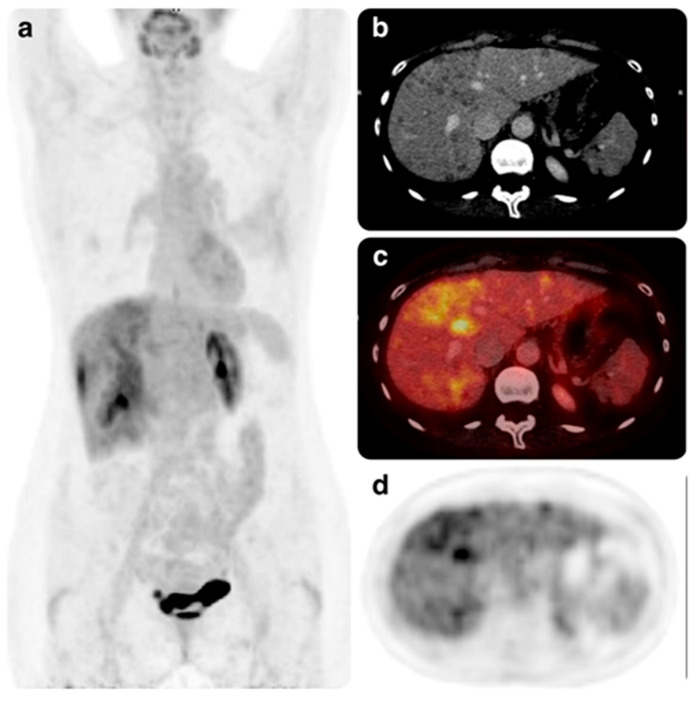
Positron emission tomography–computed tomography Scan (PET-CT) performed in July 2021 from patient 6. (**a**) Maximum intensity projection (MIP) image shows pathological fluorodeoxyglucose uptake in the liver. (**b**) CT images show multiple low-attenuation lesions in several hepatic segments, some of them confluent. (**c**,**d**) Lesions show intense fluorodeoxyglucose uptake in the fusion and PET images, respectively. These findings are compatible with multiple metastatic involvement.

**Figure 3 cancers-15-04452-f003:**
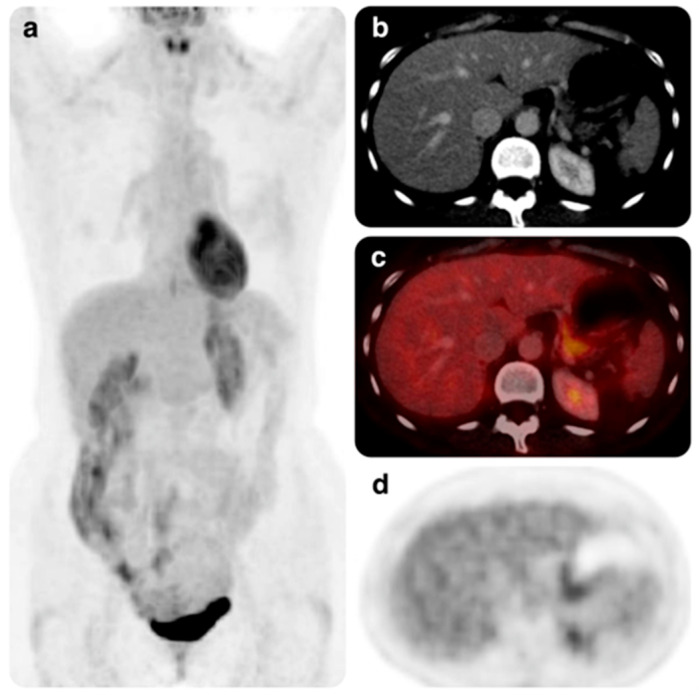
PET-CT study performed in November 2021 from patient 6. (**a**) MIP image showing the disappearance of the hepatic pathologic uptakes visible in the previous study. (**b**) CT images show no lesions. (**c**,**d**) No pathological fluorodeoxyglucose uptakes are seen in the fusion and PET images. These findings, therefore, suggest a complete response of liver metastases.

**Table 1 cancers-15-04452-t001:** Patient and disease characteristics. CDK, cyclin-dependent kinase; CNS, central nervous system.

Characteristic	Patients Included (n = 11)
Age in years, median (range)	69 (42–84)
Sex, n (%)	
Female	11 (100)
Ethnicity, n (%)	
Caucasian	10 (90.9)
Hispanic	1 (9.1)
Median number of previous lines (range)	3 (1–7)
Previous treatment modalities, n (%)	
Endocrine therapy	11 (100)
Chemoendocrine therapy	6 (54.4)
Prior CDK4/6 inhibitors	
Palbociclib	7 (63.6)
Ribociclib	4 (36.4)
Median time from metastatic diagnosis to treatment with abemaciclib in months, median (range)	50 (21–71)
Median time until treatment with abemaciclib in months since previous CDK4/6 inhibitors (range)	17.5 (3–41)
Median number of endocrine therapies until treatment with abemaciclib since previous CDK4/6 inhibitors (range)	1 (0–2)
Median number of chemotherapies until treatment with abemaciclib since previous CDK4/6 inhibitors (range)	2 (0–6)
Combined therapy, n (%)	
Monotherapy	1 (9.1)
Tamoxifen	9 (81.8)
Aromatase inhibitor	1 (9.1)
Visceral involvement, n (%)	
Yes	8 (72.7)
No	3 (27.3)

**Table 2 cancers-15-04452-t002:** Best response and PFS.

Patient ID	Prior CDK4/6 Inhibitor	Best Response	Calculated PFS (Months)
1	Ribociclib	PR	6
2	Ribociclib	PD	4
3	Palbociclib	PR	10
4	Palbociclib	PD	2
5	Palbociclib	PR	7 (ongoing)
6	Ribociclib	CR	7 (ongoing)
7	Ribociclib	SD	8
8	Palbociclib	PD	3
9	Palbociclib	PR	5 (ongoing)
10	Ribociclib	SD	2 (ongoing)
11	Palbociclib	PR	2 (ongoing)

**Table 3 cancers-15-04452-t003:** Characteristics of the studies exploring the benefit of continuing CDK4/6 inhibition after disease progression with prior CDK4/6 inhibitors.

Name of Study	Phase of Study	Drug	Prior CDK4/6 Inhibitors Used	Arms of Treatment	Primary Outcome Measures	Result
**postMONARCH** [36]	III	Abemaciclib	Palbociclib, abemaciclib, ribociclib	Abemaciclib plus fulvestrant vs. placebo plus fulvestrant	PFS	Pending
**MAINTAIN** [40]	II	Ribociclib	Palbociclib, abemaciclib, ribociclib	Ribociclib plus fulvestrant/exemestane vs. placebo plus fulvestrant/exemestane	PFS Percentage of patients free of progression at 24 weeks from the start of the study	Positive
**PACE** [41]	II	Palbociclib	Palbociclib, abemaciclib, ribociclib	Fulvestrant vs. fulvestrant with palbociclib vs. fulvestrant with palbociclib and avelumab	PFS	Negative
**PALMIRA** [42]	II	Palbociclib	Palbociclib Must have achieved clinical benefit criteria in response to a first-line palbociclib-based endocrine regimen	Palbociclib plus letrozole/fulvestrant vs. letrozole/fulvestrant	PFS	Negative

## Data Availability

The datasets generated during the current study are not publicly available but are available from the corresponding author upon reasonable request.

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
