# Peer review of "Clinical Experience with Abemaciclib in Patients Previously Treated with Another CDK 4/6 Inhibitor in a Tertiary Hospital: A Case Series Study"

_cancers, 2023, doi:10.3390/cancers15184452_

Round 1

Reviewer 1 Report

The article entitled "Clinical experience with abemaciclib in patients previously treated with another CDK 4/6 inhibitor in a tertiary hospital in Madrid, Spain" presents data that could be relevant for breast cancer patients who have received previous treatments with CDK4/6 inhibitors. Although the premise is interesting, the way the authors present the work is not very understandable, so I have the following observations.

Major observations:

The supplementary material "Data set of patients included in the study" should be presented as a table in the main article format.

The resolution of the PET-CT images needs to be improved.

The presentation of Table 3 should be improved.

Figure 2 is not mentioned in the text and should be one of the key figures and results of the study. The points on the curve should represent the actual events observed in the study. Each progression event should be shown as a downward step in the curve, and censoring periods (when patients drop out of the study before experiencing progression) should be indicated as flat points on the curve. Confidence intervals should also be indicated. It is common to show confidence intervals around the Kaplan-Meier curve to indicate the uncertainty associated with the estimation of progression-free survival. These intervals provide an idea of the precision of the estimation and are usually represented as shaded bands around the curve. A Kaplan-Meier curve for progression-free survival on abemaciclib provides a visual representation of the proportion of patients who remain free from progression over time, based on the observed events in a clinical study or clinical trial. The figure is not described.

The authors state that Abemaciclib may be an effective and safe treatment option in MBC patients previously treated with palbociclib or ribociclib, but they only show one representative case.

Minor observations:

The article format should be reviewed before submission, removing lines 53-68.

The abstract provided for review does not match the abstract in the article.

Moderate editing of English language required

Author Response

The supplementary material "Data set of patients included in the study" should be presented as a table in the main article format. Thank you so much for your comment, these data has been also included in the main article as Table 2.

The resolution of the PET-CT images needs to be improved. We really thank you for this suggestion, images have been uploaded again to improve the resolution. Image a corresponds to Maximum intensity projection (MIP), b to CT images and c to fusion images. d corresponds to PET images and its resolution cannot be improved.

The presentation of Table 3 should be improved. We appreciate this comment. Data have been updated and improved.

Figure 2 is not mentioned in the text and should be one of the key figures and results of the study. The points on the curve should represent the actual events observed in the study. Each progression event should be shown as a downward step in the curve, and censoring periods (when patients drop out of the study before experiencing progression) should be indicated as flat points on the curve. Confidence intervals should also be indicated. It is common to show confidence intervals around the Kaplan-Meier curve to indicate the uncertainty associated with the estimation of progression-free survival. These intervals provide an idea of the precision of the estimation and are usually represented as shaded bands around the curve. A Kaplan-Meier curve for progression-free survival on abemaciclib provides a visual representation of the proportion of patients who remain free from progression over time, based on the observed events in a clinical study or clinical trial. The figure is not described. Thank you for your valuable comments. We have revised the Kaplan-Meier curve for progression-free survival on abemaciclib to represent actual events as downward steps. However, as no patients dropped out of the study, there are no censoring periods. We have included confidence intervals as shaded bands around the curve to indicate the uncertainty associated with the estimations. Additionally, we have updated the figure description to highlight that the curve depicts the proportion of patients remaining free from progression based on observed events in our clinical study. Thank you for your valuable input.

The authors state that Abemaciclib may be an effective and safe treatment option in MBC patients previously treated with palbociclib or ribociclib, but they only show one representative case. We are grateful for this comment. In this article, we present data for a small number of patients. In the results we  emphasise the case of one patient that achieved a complete response, but a benefit in PFS and partial responses were achieved by other patients included in the study, as it is showed in Table 2. Considering the current data available about treatment after progression on a CDK4/6 inhibitor, we reckon that the benefit seen in our study might be relevant. 

Minor observations:

The article format should be reviewed before submission, removing lines 53-68. We appreciate this observation, these lines have been removed.

The abstract provided for review does not match the abstract in the article. Thank you for insightful comment, the correct abstract is the one that is included in the main article, that has been updated.

Reviewer 2 Report

This is an interesting and important report but due to the low number of subjects [eleven], it is more readable as a case report. I like the excellent introduction and discussion with a very good review of the literature..

Author Response

This is an interesting and important report but due to the low number of subjects [eleven], it is more readable as a case report. I like the excellent introduction and discussion with a very good review of the literature.

We would like to express our sincere gratitude for your valuable and insightful commentary. We would like to emphasise that the strategy we proposed was not a standard approach when we initially planned our study. Given this circumstance and the availability of different ongoing trials in this setting, we felt it would be in the best interest of the patients to refer them to one of the centres that were conducting the trial, so no further patients underwent this strategy in our centre. Controversial data has been published for palbociclib and ribociclib, so we consider that even the small sample of the study, this real word data would add more evidence on the efficacy of abemaciclib when the results of a randomized phase III trial have not been published yet.

Reviewer 3 Report

The authors report on their clinical experience with abemaciclib in patients previously treated with another CDK 4/6 inhibitor. It is a well written manuscript addressing an important question. The language is quite ok, some typos.

The major problem is that it is a retrospective analysis with a heterogenious collective, as stated by the authors themselves (line 174). Because of a selection bias all findings regarding efficacy abd safety are scientifically not sound. No normal distribution of the finding may be assumed. No conclusions can be drawn. The discussion is quite good as a review of literature, and the reported cases are interesting.

I recommend the authors to rearange the manuscript as a case report with a review of literature and send the manuscript to a dedicated journal for case reports.

Further remarks:

1. It is unusual to mention the name of the city in the title.

2. Simple summary: The sentence "these findings indicate ..." is completely wrong because of the arguments listed above.

3. page 2: parts of the template were not deleted.

4. Figure 1: this figure mimics a prospective trial and brings no additional value. Should be deleted.

5. Table 3: a summary of characteristics of trials. This is ok., but aren't there any results?

The language is quite ok, some typos.

Author Response

The authors report on their clinical experience with abemaciclib in patients previously treated with another CDK 4/6 inhibitor. It is a well written manuscript addressing an important question. The language is quite ok, some typos.

The major problem is that it is a retrospective analysis with a heterogeneous collective, as stated by the authors themselves (line 174). Because of a selection bias all findings regarding efficacy and safety are scientifically not sound. No normal distribution of the finding may be assumed. No conclusions can be drawn. The discussion is quite good as a review of literature, and the reported cases are interesting.

I recommend the authors to rearrange the manuscript as a case report with a review of literature and send the manuscript to a dedicated journal for case reports.

We really appreciate your comments. We agree with the limitations of the study, however we think that patients with HR positive metastatic breast cancer are a very heterogeneous group of patients with different patterns of aggressiveness in where different treatment strategies can be developed. For this, we think that these real word data could add an important value to the data from the randomized clinical trial that is currently ongoing.

Further remarks:

1. It is unusual to mention the name of the city in the title. Many thanks for this comment, it has been deleted from the title.

2. Simple summary: The sentence "these findings indicate ..." is completely wrong because of the arguments listed above. Thank you for this valuable comment, this sentence has been modified to clarify the potential value of the findings.

3. page 2: parts of the template were not deleted. Thank you, it has been deleted now.

4. Figure 1: this figure mimics a prospective trial and brings no additional value. Should be deleted. Many thanks for this suggestion. The figure has been deleted.

5. Table 3: a summary of characteristics of trials. This is ok., but aren't there any results? We are very grateful for this comment, this table has been updated and improved.

Round 2

Reviewer 1 Report

The authors have made significant improvements to the article. I still believe that the work would be better suited for a case-control study.

In the Results section, specifically in subsection 3.1, simply mentioning "patients" does not provide much information. It is important to include the most relevant characteristics of these patients, as well as in subsections 3.2 and 3.3.

The tables should be presented in a format appropriate for an article.

Minor editing of English language required, Still typos

Author Response

The authors have made significant improvements to the article. I still believe that the work would be better suited for a case-control study. Your positive feedback is sincerely appreciated. Regarding the suggestion that our study might be better suited for a case-control design, we would like to clarify that our article is a retrospective cohort study, and this study design was deliberately chosen to address our research objectives. Given the availability of data, a retrospective cohort study was the most suitable approach for our investigation. We express our gratitude for your insightful comments.

In the Results section, specifically in subsection 3.1, simply mentioning "patients" does not provide much information. It is important to include the most relevant characteristics of these patients, as well as in subsections 3.2 and 3.3. We sincerely thank you reviewer for providing valuable feedback. In the Results section, specifically in subsection 3.1, we included some relevant characteristics as median age, median time to abemaciclib from the end of previous CDK4/6 inhibitors, median number of therapies until abemaciclib use, disease extension and the combined therapy. Now we have included additional information about the patients' ethnicity. This addition will provide readers with a clearer understanding of the study population. Regarding subsections 3.2 and 3.3, we want to highlight that we have presented the most relevant characteristics of the patients in the respective tables and figures accompanying the text. By referring to these tables and figures, readers can gain a more comprehensive and in-depth insight into the patient characteristics in relation to the findings.

The tables should be presented in a format appropriate for an article. We sincerely appreciate the time and effort you have dedicated to reviewing our manuscript. Regarding the presentation of tables, we want to assure you that the format in which the tables are presented follows the recommendations provided by the journal. We have adhered to the journal's specific formatting requirements to ensure consistency and compatibility with the publication standards. Once again, we express our gratitude for your thoughtful comments, which have been instrumental in improving the quality of our manuscript. 

Reviewer 3 Report

Some improvements were made, the major limitations of the manuscript remain unchanged:

'The major problem is that it is a retrospective analysis with a heterogeneous collective, as stated by the authors themselves (line 174). Because of a selection bias all findings regarding efficacy and safety are scientifically not sound. No normal distribution of the finding may be assumed. No conclusions can be drawn. The discussion is quite good as a review of literature, and the reported cases are interesting.'

No further comments

Author Response

We really appreciate this comment. We are aware about the limitations of the study so we have focused the discussion on a literature review and the aim to present real-world data in this setting. Once again we really appreciate your insightful feedback.